# A Flexible Piezocapacitive Pressure Sensor with Microsphere-Array Electrodes

**DOI:** 10.3390/nano13111702

**Published:** 2023-05-23

**Authors:** Shu Ying, Jiean Li, Jinrong Huang, Jia-Han Zhang, Jing Zhang, Yongchang Jiang, Xidi Sun, Lijia Pan, Yi Shi

**Affiliations:** Collaborative Innovation Center of Advanced Microstructures, School of Electronic Science and Engineering, Nanjing University, Nanjing 210093, China; yingshu@smail.nju.edu.cn (S.Y.); jali@smail.nju.edu.cn (J.L.); jhzhang@smail.nju.edu.cn (J.-H.Z.); zjing@smail.nju.edu.cn (J.Z.); ycjiang@nju.edu.cn (Y.J.); yshi@nju.edu.cn (Y.S.)

**Keywords:** microsphere arrays, nanofiber dielectric layers, piezocapacitive sensor, flexible pressure sensors, electronic skins

## Abstract

Flexible pressure sensors that emulate the sensation and characteristics of natural skins are of great importance in wearable medical devices, intelligent robots, and human–machine interfaces. The microstructure of the pressure-sensitive layer plays a significant role in the sensor’s overall performance. However, microstructures usually require complex and costly processes such as photolithography or chemical etching for fabrication. This paper proposes a novel approach that combines self-assembled technology to prepare a high-performance flexible capacitive pressure sensor with a microsphere-array gold electrode and a nanofiber nonwoven dielectric material. When subjected to pressure, the microsphere structures of the gold electrode deform via compressing the medium layer, leading to a significant increase in the relative area between the electrodes and a corresponding change in the thickness of the medium layer, as simulated in COMSOL simulations and experiments, which presents high sensitivity (1.807 kPa^−1^). The developed sensor demonstrates excellent performance in detecting signals such as slight object deformations and human finger bending.

## 1. Introduction

In contemporary society, the burgeoning Internet of Things (IoT) technology has led to an upsurge in wearable devices, which continuously monitor the human body while interacting with the terminal, and sensors play a crucial role in this regard [1,2,3,4,5]. Human skin, being the largest and most basic organ of the human body, serves as both the primary barrier to the external environment and a source of sensory information regarding external stimuli such as pressure, deformation, and touch [6,7]. Hence, electronic skin has become essential in wearable medical devices, human–computer interaction, artificial intelligence, and health monitoring [8,9,10,11,12,13,14,15]. An ideal e-skin must possess high flexibility, sensitivity, affordability, and structural simplicity and should mimic the natural skin’s ability to sense tactile pressure ranging from light contact to object haptic perception (0–30 kPa) [16,17,18,19]. Piezoresistive, piezocapacitive, triboelectric, and piezoelectric sensing mechanisms are commonly employed to convert the applied stress signal into an electrical signal in electronic skin pressure sensors. In particular, piezocapacitive pressure sensors are highly attractive owing to their ease of preparation, structural simplicity, low power consumption, and compact circuit layout [20,21,22,23,24,25,26,27,28,29,30].

The sensing performance of piezocapacitive sensors primarily depends on the mechanical properties of the elastic dielectric layer used. In some respects, the greater the compressibility under the same pressure of the dielectric material used, the higher the sensitivity of the sensor [15,20]. Capacitive pressure sensors usually consist of two parallel electrodes sandwiched on a polymer dielectric layer. Since the sensor’s capacitance changes by altering the thickness of the dielectric layer when an external force is applied to the sensor, its sensitivity is directly affected by the mechanical properties of the elastic dielectric layer and the overall thickness of the sensor. Therefore, polydimethylsiloxane (PDMS) with high compressibility is considered a suitable material for pressure-sensitive dielectric films [31,32,33]. By modifying the microstructure of PDMS, the surface of the pressure-sensitive layer can be elastically deformed, and it can quickly recover its original shape upon the application/release of external pressure, in which the microstructure improves the overall sensor performance and mitigates issues associated with viscoelastic behavior. To further enhance the overall performance of piezocapacitive pressure sensors—measured in terms of linearity, sensitivity, detection range, and stability—microstructures such as pyramidal structures, micropores, and air gaps are often incorporated into the dielectric layer to obtain an appropriate Young’s modulus or silver nanowire embedding is introduced to increase the dielectric constant [14,34,35]. However, these preparation methods typically employ porous structures to enhance the sensor’s sensitivity, making it challenging to achieve high sensitivity, wide detection range, and good linearity simultaneously [12,33,36,37,38]. Moreover, these microstructures necessitate complex processes such as photolithography, sacrificial templates, or chemical etching, which are unsuitable for low-cost and large-scale production. Hence, there is an urgent need to develop a simple and reliable preparation method that can customize the microstructures easily, thus enhancing the sensor’s performance while reducing the preparation cost.

In this paper, we propose a low-cost, high-sensitivity, and large-area compatible inverse-mold technique for preparing a flexible capacitive pressure sensor with a self-assembled microsphere-type array. We simulate the designed sensor using COMSOL, including the simulation of the nonwoven fabrics with electrostatic spinning and the calculation of the overall sensor performance, to provide a theoretical basis for the experiments. With the help of COMSOL simulations, we prepared a highly sensitive piezocapacitive sensor, which exhibited a relative rate of change ΔC/C_0_ maximum of about 2.13, a sensitivity of 1.807 kPa^−1^ maximum, and a hysteresis of 7.69%. These performance improvements were attributed to the interaction between the microsphere-shaped gold electrode array and the rough surface of the dielectric layer, resulting in a significant change in the contact area of the electrodes and the thickness of the dielectric layer. Moreover, the electrostatic spinning method used to prepare the nanofibrous dielectric layer makes it more easily compressible compared to a conventional dielectric layer. The exclusion of air from the dielectric layer also increases its dielectric constant, making the change in capacitance more pronounced. Consequently, this flexible pressure sensor exhibits excellent performance in pressure measurement and has significant potential for electronic skin applications. Overall, the results of our study suggest that the proposed low-cost and high-performance capacitive pressure sensor could find wide applications in the fields of robotics, prosthetics, and healthcare monitoring.

## 2. Materials and Methods

### 2.1. COMSOL Multiphysics Simulation

We performed simulations based on the focuses of two modules within COMSOL Multiphysics: electrostatics and solid mechanics. To obtain the potential map of a capacitive pressure sensor, we first used the electrostatics sub-module under the electromagnetics module to calculate the electric field generated by electrostatics. Under free space, the spatial electric field is irrotational, and the relationship between the potential and the spatial charge density can be determined using Maxwell’s system of equations:(1)$$-\nabla \cdot \nabla V=\frac{\rho }{\varepsilon_{0}}$$
where *ρ* is the charge density, *ε*_0_ is the free dielectric constant in space, and *V* is the electric potential. However, the internal electric field of a dielectric material is generated by an electric dipole, which is not consistent with the free electric field. Therefore, this phenomenon needs to be described from a macroscopic point of view:(2)$$\rho_{P}=-\nabla \cdot P$$
(3)$$D=\varepsilon_{0}E+P$$
where *P* is the polarization vector field and *D* is the electrical displacement density. The set of electrostatic equations can be finally associated as a system of equations:(4)$$-\nabla \cdot (\varepsilon_{0}\nabla V-P)=\rho_{P}$$

When a material is compressed or stretched, the stress distribution within it is not uniform. Therefore, our simulation needed to consider the physical properties of the material, the stress distribution, and the deformation. In solid mechanics, the stress tensor of a material is used to describe the stress distribution in three dimensions:(5)$$\sigma =\left(\begin{array}{l} \sigma_{xx}\;\sigma_{xy}\;\sigma_{xz}\\ \sigma_{yx}\;\sigma_{yy}\;\sigma_{yz}\\ \sigma_{zx}\;\sigma_{zy}\;\sigma_{zz} \end{array}\right)$$
where the first component represents the component of the force and the second represents the direction of the normal to the force. Newton’s second law can be rewritten to be expressed using the stress tensor:(6)$$\nabla \cdot \sigma +f=\rho \frac{\partial^{2}u}{\partial t^{2}}$$
where *f* is the force per unit volume, *ρ* is the mass density, and *u* is the displacement vector. When a material is subjected to stress, it deforms, and stresses are generated within it. When deformation occurs, all parts of the material should be in harmony and reach a state of equilibrium. In order to describe this strain, a stacking tensor element is introduced to represent it:(7)$$\varepsilon =\left(\begin{array}{l} \varepsilon_{xx}\;\varepsilon_{xy}\;\varepsilon_{xz}\\ \varepsilon_{yx}\;\varepsilon_{yy}\;\varepsilon_{yz}\\ \varepsilon_{zx}\;\varepsilon_{zy}\;\varepsilon_{zz} \end{array}\right)$$

Each tensor element is defined as the derivative of the displacement:(8)$$\left[\begin{array}{l} \varepsilon_{xx}\\ \varepsilon_{yy}\\ \varepsilon_{zz}\\ \varepsilon_{xy}\\ \varepsilon_{yz}\\ \varepsilon_{xz} \end{array}\right]=\left[\begin{array}{c} \frac{\partial u}{\partial x}\\ \frac{\partial v}{\partial y}\\ \frac{\partial w}{\partial z}\\ \frac{1}{2}\left(\frac{\partial u}{\partial y}+\frac{\partial v}{\partial x}\right)\\ \frac{1}{2}\left(\frac{\partial v}{\partial z}+\frac{\partial w}{\partial y}\right)\\ \frac{1}{2}\left(\frac{\partial u}{\partial z}+\frac{\partial w}{\partial x}\right) \end{array}\right]$$

These tensors are not characterized by an arbitrary spatial distribution of displacements, but rather by fundamental constants that can represent deformation processes and provide coordination relations for the deformation of materials under the action of external forces.

### 2.2. Chemicals and Materials

PDMS (Dow Corning, Sylgard 184) and polystyrene microspheres (PS, dm 0.6–1.0 μm, 2.5% *w*/*v*) were purchased as received. Deionized water (water purifier), ethanol solution (AR, Nanjing Reagent), gold target (ZhongNuo Advanced Material Technology Co., Ltd., Beijing, China), glass slides (10 cm diameter), and acetone solution (Shanghai Reagent) were purchased from the reagent platform of Nanjing University.

To prepare the nanofiber PVDF nonwoven fabrics, 5 mL of DMF and 5 mL of acetone solution were mixed in a beaker at a ratio of 1:1 and placed in a magnetic stirrer at 60 °C for 20 min. A certain amount of PVDF powder was added to the DMF and acetone solution to form a mixture with a mass fraction of 20% wt. The stirred mixture was then placed in a magnetic stirrer at 70 °C for 1 h. The stirred mixture was left at room temperature for 1 h to remove any air bubbles. The mixed PVDF solution was electrostatically spun at 20 kV for 6 h to obtain the nanofiber PVDF nonwoven fabrics with electrostatic spinning.

### 2.3. Preparation of Microsphere Array Electrodes

The PDMS was obtained by mixing the crosslinker with the prepolymer in the ratio of 1:10 and stirring well. It was then put into a constant-temperature desiccator for the first evacuation operation to remove the air bubbles generated during the mixing process. The PDMS was uniformly poured on the glass sheet with PS microspheres, and then the second evacuation was performed to remove the air bubbles generated during the pouring process. The glass sheet with PDMS was then placed in a constant-temperature drying oven and baked at 100 °C for 2 h to cure and form a smooth PDMS film. The PDMS substrate with inverted microsphere arrays can be obtained by tearing off the PDMS film. The PDMS substrate was washed in acetone solution to remove the residual PS microspheres, then cleaned with ethanol and deionized water, respectively, and placed on a hot table for drying. The PDMS substrates with inverted microsphere arrays obtained from the first inversion were placed in a UV-Ozone cleaning machine for 20 min for hydrophilic treatment to produce a surface with good hydrophilicity. The second PDMS layer was uniformly poured onto the PDMS surface of the first layer, placed in an oven and baked at 80 °C for 2 h, and then removed. The PDMS substrate with microsphere arrays was prepared by the “secondary inversion” method by carefully tearing off the PDMS film of the second layer. Finally, the final PDMS substrate was sputtered with gold for 5 min using a JS-1600 magnetron sputterer to obtain gold electrode arrays of 50 nm thickness.

### 2.4. Preparation of Pressure Sensors

Copper wires and gold electrodes with a microsphere array structure were bonded together with conductive silver paste. Then, the lower electrode, nonwoven fabrics with electrostatic spinning, and upper electrode were assembled together in a sandwich structure to obtain a flexible capacitive pressure sensor.

## 3. Results and Discussion

Figure 1a depicts the fabrication process of microsphere array electrodes (Appendix A) for flexible capacitive pressure sensors. The process begins with the dispersion of PS microspheres in an ethanol solution, followed by the uniform deposition of the solution on a glass sheet to form a monolayer film through self-assembly. Once the ethanol evaporates completely, the PS microspheres are evenly distributed on the monolayer glass sheet, as shown in Figure 1b. Next, the configured PDMS solution is poured onto the PS microsphere film to form a concave microsphere structure after curing. Subsequently, PDMS is poured onto the concave PDMS film formed for the first time by the second inversion method and cured in the same way to form a PDMS substrate with a microsphere array structure, as shown in Figure 1c. Finally, the microsphere gold electrode arrays, with a gold layer of 50 nm in thickness, are obtained through magnetron sputtering.

In the process of preparing flexible capacitive pressure sensors, electrostatic spinning plays a crucial role. Due to the complex network structure of the nanofiber nonwoven fabrics, the electrostatic spinning time needs to be controlled for more than six hours to prevent any leakage current or a device short-circuit. The electrostatic spinning method is used to prepare the PVDF film, as shown in Figure 1d. A dielectric layer with fibers exhibits a higher displacement increase compared to a flat dielectric layer without fibers. The nanofibers with voids constitute the inner part of the dielectric layer prepared by the electrostatic spinning technique. The interlaced nanofibers can be easily deformed by applying external forces, which is advantageous in detecting small force changes. From the SEM photos in the inset of Figure 1d, it can be seen that the diameter of PVDF nanofibers of the nonwoven fabrics is about 200 nm, the thickness is relatively uniform and smooth, the boundary is clear and uniformly distributed, and there is practically no agglomeration phenomenon.

The performance of capacitive pressure sensors can be improved through altering the electrode contact area and dielectric layer parameters. We first simulated the pressure sensor model with a microsphere-array-structure gold electrode on the surface and a fibrous dielectric material layer with surface roughness produced via electrostatic spinning technology through COMSOL, as shown in Figure 2a. Since the nanofibrous dielectric layer itself has a very complex fiber structure on the surface and inside, it is not feasible to make a complete fibrous dielectric layer. Thus, only rectangular voids can be added in the middle of the model of the nanofibrous dielectric layer to simulate the complex fiber structure. Figure 2b shows the deformation of the nonwoven fabrics with electrostatic spinning in the pressure range of 0–20 kPa. As the thickness of the dielectric layer decreases, its contact area with the motor increases gradually, resulting in an increase in capacitance value with the increase of pressure.

Subsequently, the simulation model without the nanofibrous structured dielectric layer is selected for comparison experiments. Figure 2c shows that the displacement increase of the dielectric layer with the fibrous form is significantly higher than that without the fibrous form. When P_0_ is 2 kPa, the displacement with the fibrous dielectric layer reaches about 0.57 μm, while the displacement without the fibrous dielectric layer is only 0.40 μm.

In a flexible capacitive pressure sensor utilizing a microsphere array, the electrode portion is formed by sputtering a metal layer onto a PDMS surface with an array of microspheres, and this electrode design can effectively improve the sensitivity of the device (Appendix A). The application of pressure onto the substrate portion of the sensor results in the deformation of the electrodes as they come into contact with the dielectric layer, as illustrated in Figure 3a. The gold microspheres, with a thickness of only 100 nm, cause the electrodes to be squeezed and their contact area to change, ultimately leading to a change in the capacitance of the sensor. In contrast to the parallel plate electrode, the electrode area of the capacitive pressure sensor undergoes a greater degree of change, resulting in a greater amount of capacitance change. This paper presents a flexible capacitive pressure sensor model designed to account for changes in both the dielectric layer and electrode, as shown in Figure 3b. Based on this model, the theoretical sensitivity achievable by the sensor is calculated. Due to the large calculation volume of the three-dimensional model, a two-dimensional model calculation was utilized to maintain consistent imposed conditions. While an ordinary dielectric layer is used in this paper instead of a nanofibrous dielectric layer, parameters such as the elastic modulus and Poisson’s ratio of the nanofibrous dielectric layer are introduced to prevent the experimental results from being affected. Additionally, the microsphere-structure gold electrode is known to deform under applied pressure, which is difficult to simulate; thus, this paper employs a microsphere structure that does not deform instead. Figure 3c displays the potential diagram of the electric field following pressure application. The thickness of the dielectric layer decreases under the action of pressure P0, resulting in an increase in the contact area of the electrode and, consequently, an increase in the capacitance of the entire capacitor. The initial capacitance (C_0_) is 1.51 pF when no pressure is applied. As pressure P0 increases, the capacitance also increases and ultimately reaches approximately 6.4 pF, as illustrated in Figure 3d. The graph in Figure 3e illustrates the relative rate of change of capacitance ΔC/C_0_ for the simulation model, with a maximum value of about 3.238 observed when the pressure reaches a maximum of 20 kPa.

The performance variation curve plotted from the simulation data shows that the nanofibrous dielectric layer has good deformation capability, while the sensor has excellent sensing performance under small stress. We bonded copper wires and gold electrodes with microsphere arrays together using a conductive silver paste/glue. Then, we assembled the lower electrode, the nonwoven fabrics with electrostatic spinning, and the upper electrode according to the “sandwich” structure to produce a flexible capacitive pressure sensor based on the microsphere array. In order to obtain high sensitivity, the dielectric layer thickness of the device needs to be controlled within 5 μm, so the test voltage needs to be controlled at 1 V to prevent the device from breakdown during testing. Figure 4a shows the capacitance pressure curve of the sensor, which shows that the capacitance value of the sensor increases with increasing pressure, increasing to 27.90 pF when the pressure reaches 80 kPa. The sensitivity of the sensor is very large at stresses less than 1 kPa. Similar results can be obtained for the variation curve of the relative rate of change of capacitance’s relative intensity, as shown in Figure 4b, which can reach a value of 2.13 when the pressure reaches 80 kPa. The sensitivity curve in Figure 4c shows that the decrease in sensitivity is very sharp when the pressure is less than 1 kPa, and the sensitivity can reach a maximum of 1.807 kPa^−1^ at 0.05 kPa. This is due to the fact that when the electrode is subjected to a small pressure, the surface of the dielectric layer is squeezed against the electrode, resulting in a change in contact area, which can lead to a dramatic change in the capacitance value of the sensor, making the sensor excellent for small force measurements. Moreover, when more pressure is applied to the sensor, the thickness of the dielectric layer decreases further, causing an increase in its capacitance value. This property is related to the internal structure of the dielectric layer prepared by electrostatic spinning, which comprises nanofibers filled with air. When pressure is applied to the sensor, the air within its internal dielectric layer is compressed, causing the dielectric layer’s structure to become very dense and the dielectric constant to change accordingly, leading to an increase in the capacitance value of the sensor. This is one of the primary reasons for the change in the capacitance value of flexible capacitive pressure sensors. The capacitance test curves of the sensor are generally consistent with the simulated expectations but show significant deviations in the low-stress range. This is primarily due to the intricate fibers in the nonwoven fabrics with electrostatic spinning, where the nanofibers undergo a significant change in dielectric constant under small stresses, resulting in a sharp increase in capacitance. Treating the microscopic spheres as non-deformable objects in the simulation also leads to significant deviations between the real and simulated results in the low-pressure section. Therefore, we can observe that the performance of flexible capacitive pressure sensors can be improved through changing the microstructure of the elastomer or the surface roughness and internal structure of the dielectric layer. Figure 4d illustrates the hysteresis performance of the sensor. The process of increasing and then decreasing the external pressure from 0–80 kPa back to 0 kPa is represented by two curves. The red curve shows the response when the pressure increases, while the black curve shows the response when the pressure decreases. The hysteresis of the flexible capacitive pressure sensor is approximately 7.69%, which is excellent and generally satisfies the hysteresis performance requirements of general flexible sensors. However, due to viscoelastic deformation within the device, the contact area and dielectric layer of the electrode do not quickly return to their original size and shape when the pressure decreases, leading to a difference in capacitance value compared to when the pressure increases. Furthermore, the air inside the nanofibers does not immediately return, indirectly affecting the dielectric constant of the dielectric layer. Additionally, as the substrate of the sensor is made of PDMS and the dielectric layer is prepared by electrostatic spinning technology, there is some viscosity between the contact surfaces, leading to a small adhesion phenomenon during the process of pressure increase and decrease and resulting in the response lag of the sensor. Figure 4e shows the recovery and response time of the device; the device has a fast response time of 0.048 s and a fast recovery time of 0.072 s. The cycling stability of the device is shown in Figure 4f. We have carried out 1000 cycles of the device under the same 15 N stress and can see that the device has high cycling stability.

We designed several experiments to demonstrate the high sensitivity of our prepared microsphere-array electrode capacitive pressure sensor over a wide pressure range. First, we attached the device to a nitrile glove. As shown in Figure 5a, with the passage of nitrogen into the glove, our sensor can recognize the different deformations of the glove. The illustration in Figure 5a shows an example of the sensor on the re-glove, showing that the measured capacitance changes with the magnitude of the glove expansion. Next, we attached the sensor to the finger to distinguish the degree of finger flexion. As shown in Figure 5b, the capacitance value of the device changes as the degree of finger bending is changed, with the greater the bending, the greater the capacitance. The inset shows the capacitance cycling curve of the sensor under repeated finger bending extension, demonstrating the excellent durability and stability of the device (Appendix A).

## 4. Discussion

In this paper, gold electrodes with microsphere arrays on the surface were prepared using the methods of “self-assembly” and “secondary inversion”, and the dielectric layer materials with nanofiber nonwoven fabric structure were prepared using electrostatic spinning technology. The sensor performance based on the microsphere-array electrode prepared in this paper shows a high sensitivity under low stress, with a maximum relative change rate ΔC/C_0_ of 2.13, a maximum sensitivity of 1.807 kPa^−1^, and a hysteresis of 7.69%. This is attributed to the microsphere-shaped gold electrode array and the rough-surface dielectric layer as well as the nanofiber-like internal structure. We developed and experimentally validated the COMSOL model, which successfully revealed the sensing mechanism of the microsphere-array electrodes and nanofibrous dielectric. Finally, we demonstrated the sensitivity of this capacitive sensor by detecting subtle pressure changes under various experimental conditions. The excellent sensing range and ultra-high sensitivity make the sensor promising for many other potential applications in human haptic sensing, soft robotics, prosthetics, and surgical e-skin.

## Figures and Tables

**Figure 1 nanomaterials-13-01702-f001:**
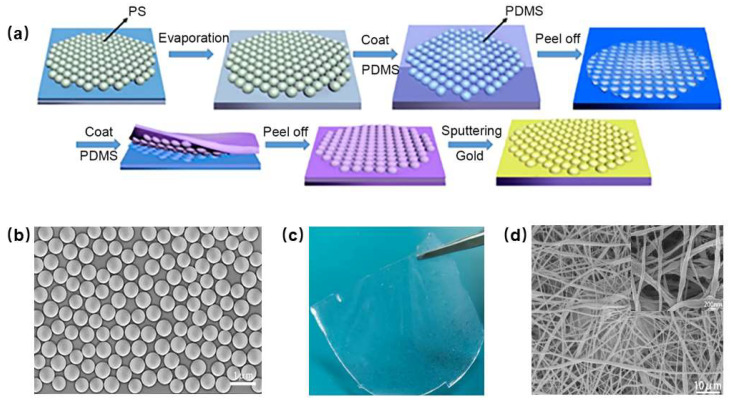
(**a**) Preparation process of microsphere array electrodes. (**b**) SEM image of monolayer PS microsphere array. (**c**) Photograph of PDMS with surface microsphere structure. (**d**) SEM image of nanofiber PVDF nonwoven fabrics; inset is SEM image at high magnification.

**Figure 2 nanomaterials-13-01702-f002:**
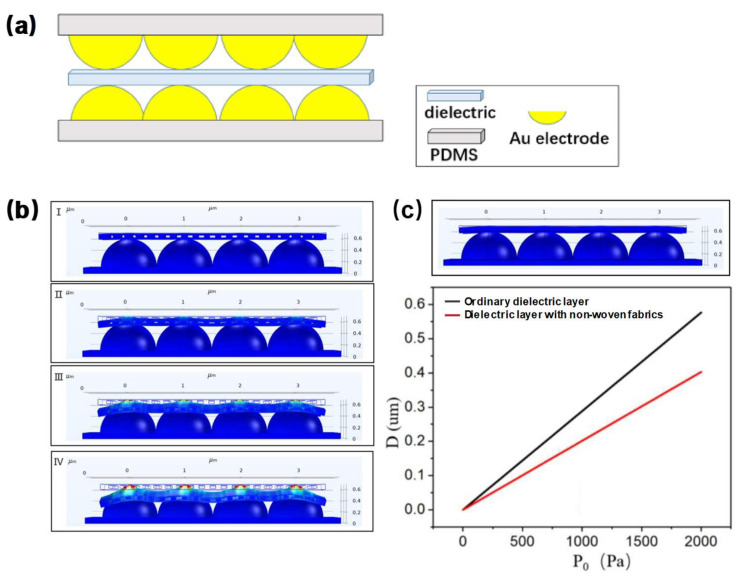
(**a**) Model drawing of flexible capacitive sensor. (**b**) The simulated deformation of the dielectric layer under the pressure of 0, 5, 10, and 20 kPa, respectively. (**c**) The upper figure shows the model of ordinary dielectric layer, and the lower figure shows the comparison of displacement of nonwoven fabrics with electrostatic spinning and ordinary dielectric layer.

**Figure 3 nanomaterials-13-01702-f003:**
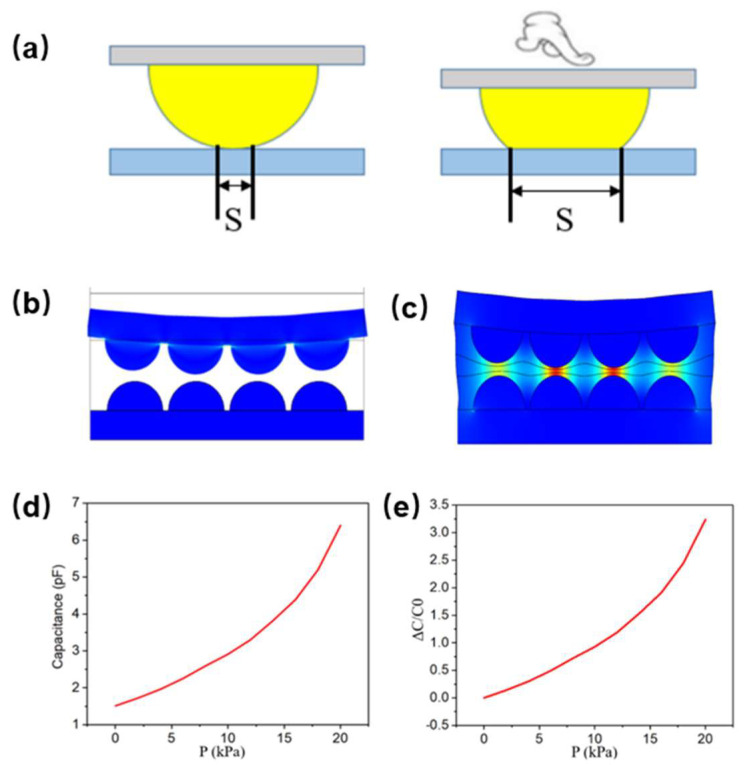
(**a**) Initial model of gold electrode with microsphere structure (**left**) and deformation under pressure (**right**). (**b**) Two−dimensional simulation model of flexible capacitive pressure sensor based on nonwoven fabrics with electrostatic spinning and microsphere−array electrode. (**c**) Simulated potential diagram of a flexible capacitive pressure sensor under external action. (**d**) Simulation curve of capacitance pressure of flexible capacitive pressure sensor. (**e**) Simulation graph of flexible capacitive pressure sensor ΔC/C_0_−pressure.

**Figure 4 nanomaterials-13-01702-f004:**
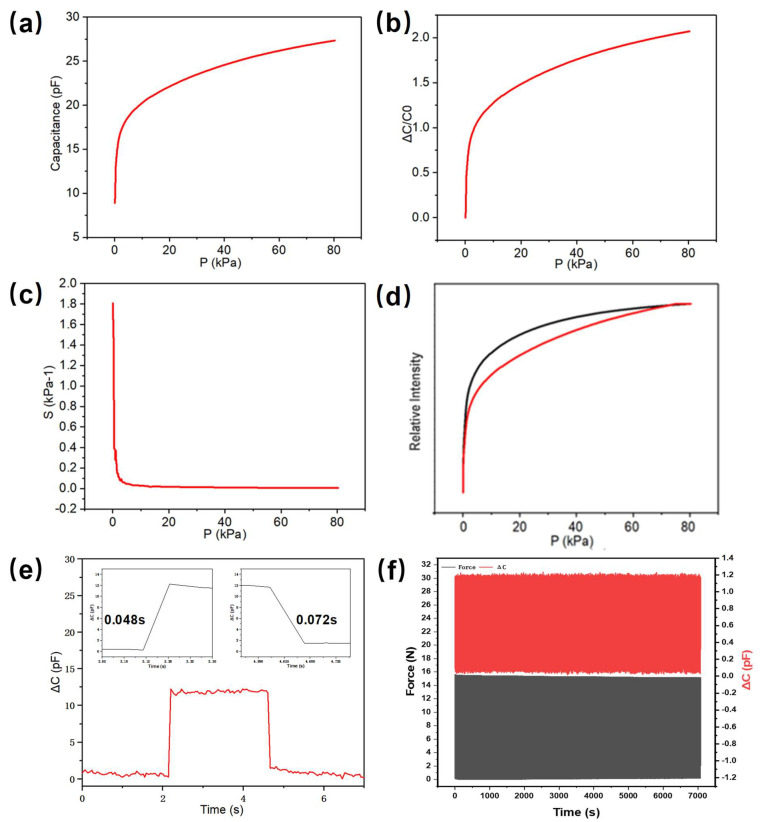
Capacitive pressure sensor based on PVDF nonwoven dielectric layer. (**a**) Capacitance versus pressure curve. (**b**) Relative rate of change of capacitance’s relative intensity versus pressure curve. (**c**) Sensitivity pressure response curve. (**d**) Capacitance single−cycle curve. (**e**) Response time. (**f**) Cycle stability.

**Figure 5 nanomaterials-13-01702-f005:**
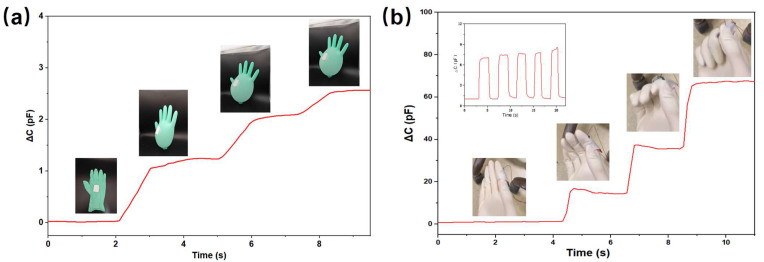
Monitoring of capacitance changes caused by various pressure sources: (**a**) different expansion sizes of the inflated gloves; (**b**) finger bending.

## Data Availability

The data presented in this study are available upon request from the corresponding author.

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
