# Peer review of "A Flexible Piezocapacitive Pressure Sensor with Microsphere-Array Electrodes"

_nanomaterials, 2023, doi:10.3390/nano13111702_

Round 1

Reviewer 1 Report

The authors report on a unique and novel flexible piezocapacitive pressure sensor with microsphere array electrodes. The study involved the preparation of gold electrodes with microsphere arrays using self-assembly and secondary inversion techniques. Additionally, dielectric layer materials with a nanofiber nonwoven fabric structure were prepared using electrostatic spinning technology. The resulting capacitive sensor showed high sensitivity under low stress, with a maximum sensitivity of 1.807 kPa-1, a maximum relative change rate of ΔC/C0 of 2.13, and a hysteresis of 7.69%. This was attributed to the microsphere-shaped gold electrode array, rough surface dielectric layer, and nanofiber-like internal structure. The researchers also developed and validated a COMSOL model, which revealed the sensing mechanism of the microsphere array electrodes and nanofibrous dielectric. Finally, the sensor was successfully used to detect subtle pressure changes under various experimental conditions, indicating potential applications in areas such as human haptic sensing, soft robotics, prosthetics, and surgical e-skin. The study presents a concise and informative introduction with references to relevant and recent literature, and the results are significant and promising for a broad scientific readership.

For the reasons mentioned above, I suggest this work deserves publication in nanomaterials, provided the following (major) comments and suggestions are fulfilled:

1.       Figure 1a shows the preparation process of microsphere array electrodes. How did the authors control the uniform coating of PS beads over the substrate? A cross-sectional SEM image of the PS beads coated film should be provided to discuss the uniformity and morphology along with SEM image provided in Figure 1b.

2.       A gold layer was sputtered over the PS beads film over the PDMS. Under strain condition, did the authors observe cracking in the film? How would this kind of deformation affect the performance of the pressure sensor/ Please comment and discuss.

3.       A cross-sectional SEM image of the pressure sensor device must be provided to confirm and validate the layering in the sandwich type pressure sensor device.

4.       Figure 5 shows the monitoring of capacitance changes caused by various pressure sources. The authors must discuss the repeatability measurements at each inflated size and finger bending condition to demonstrate the device durability and stability.

Minor editing of English language required.

Reviewer 2 Report

This is an interesting research about the development of a flexible piezocapacitive pressure sensor based on PDMS microsphere electrodes and nanofiber PVDF nonwoven fabrics. I think it can be accepted for publication after a major revision.

1. In the preparation process (Figure 1a), the authors didnot present how to deposite PVDF nanofibers on the PDMS substrate. Please add it into the figure.

2. The PS spheres in this concept were not well alligned. It is possible to align these sphere to optimize the sensor performance.

3. The detail mechanism of the enhanced sensing performace related to integration of PDMS microsphere electrodes was not explained clearly

No comment

Reviewer 3 Report

Nice paper amost ready to accept. Minor revisions as detailed in the attached file.

English almost OK. Minor corrections at the editorial stage.

Round 2

Reviewer 1 Report

The authors have carefully addressed the reviewers' comments. The manuscript can now be accepted in present form. 

Minor editing of English language required. 

Reviewer 2 Report

It can be accepted for publication

No comment